# ONETRACKERV2: UNIFIED MULTIMODAL VISUAL OBJECT TRACKING WITH MIXTURE OF EXPERTS

## ABSTRACT

Multimodal visual object tracking can be divided into to several kinds of tasks (e.g. RGB and RGB+X tracking), based on the input modality. Existing methods often train separate models for each modality or rely on pretrained models to adapt to new modalities, which limits efficiency, scalability, and usability. Thus, we introduce *OneTrackerV2*, a unified multi-modal tracking framework that enables end-to-end training for any modality. We propose Meta Merger to embed multi-modal information into a unified space, allowing flexible modality fusion and improved robustness to corrupted modalities. We further introduce Dual Mixture-of-Experts (DMoE): T-MoE models spatio-temporal relations for tracking, while M-MoE embeds multi-modal knowledge, disentangling cross-modal dependencies and reducing feature conflicts. With a shared architecture, unified parameters, and a single end-to-end training, *OneTrackerV2* achieves state-of-the-art performance across five RGB and RGB+X tracking tasks and 12 benchmarks, while maintaining high inference efficiency. Notably, even after model compression, *OneTrackerV2* retains strong performance. Moreover, *OneTrackerV2* demonstrates remarkable robustness under modality-missing scenarios.

## 1 INTRODUCTION

Visual object tracking (Zhou et al., 2023a;b; Bertinetto et al., 2016; Chen et al., 2020; Li et al., 2019a; Wu et al., 2013; Ye et al., 2022; Chen et al., 2021) aims to localize the target object in each subsequent frame of a video based on its appearance in the first frame. Depending on the input modalities, visual object tracking can be categorized into several types (Hong et al., 2024b), such as RGB tracking (where only RGB images are available) (Chen et al., 2023b;a; Wei et al., 2023; Bai et al., 2024; Zhou et al., 2025) and RGB+X tracking (where additional modalities are incorporated alongside RGB images). Furthermore, RGB+X tracking can be further divided into RGB+D (Depth) tracking (Yan et al., 2021b; Kristan et al., 2022), RGB+T (Thermal) tracking (Li et al., 2021; 2019b), RGB+E (Event) tracking (Wang et al., 2023), and RGB+N (Language) tracking (Li et al., 2017; Fan et al., 2019; Wang et al., 2021).

As illustrated in Figure 1, existing multimodal trackers can be categorized into two categories: (a) separated trackers, which design and train task-specific architectures for each modality; and (b) adaptation-based trackers such as OneTracker (Hong et al., 2024b), which fine-tune pretrained RGB trackers for downstream multimodal tasks. Despite their success, both categories of multimodal trackers suffer from several limitations. (1) **Multi-step training**, where transferring pretrained models leads to suboptimal convergence(Zhu et al., 2023; Hong et al., 2024b; Hou et al., 2024); (2) **Lack of a unified architecture**, requiring handcrafted, task-specific designs; (3) **Un-unified Parameters**, where even shared architectures (Hong et al., 2024b) rely on task-dependent weights; and (4) **Vulnerability to missing modalities**, caused by heavy reliance on fixed input configurations.

To address these issues, we advocate for a unified training paradigm that enables a single tracker to handle diverse modalities in a scalable and robust manner. Instead of designing separate architectures or relying on multi-stage fine-tuning, our approach emphasizes training from the ground up with a shared backbone, consistent parameterization, and modality-robust fusion. A key component of this framework is Meta Merger, which embeds all modalities into a joint representation space and facilitates flexible, learnable interactions between RGB and auxiliary modalities. Unlike prior methods that either double computation with separate branches or apply naive concatenation, our

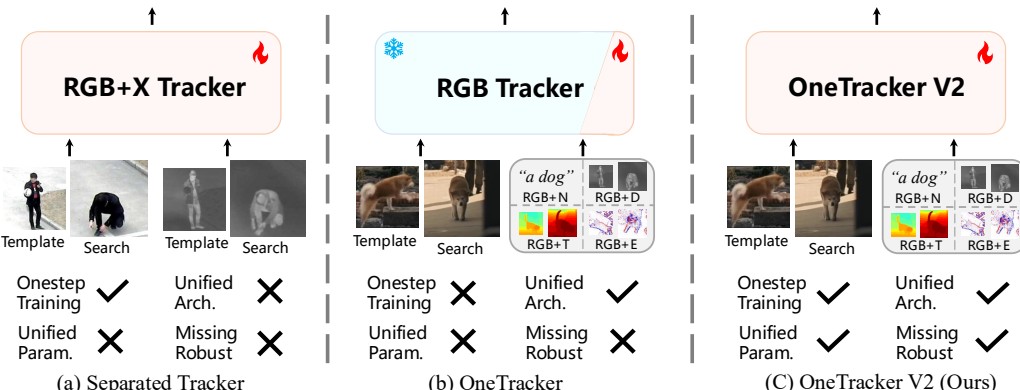

Figure 1: **Comparison of our OneTracker V2 and previous models.** (a) **Separated trackers**: task-specific architectures trained independently for each task. (b) **Fine-tuned trackers**: represented by OneTracker (Hong et al., 2024b), which adapts pretrained RGB trackers to downstream RGB+X tasks through fine-tuning. (c) *OneTrackerV2* **(Ours)**: a unified architecture with shared parameters, trained once to handle multiple multimodal tracking tasks.

unified design achieves: (1) **comprehensive cross-modal interaction**, where the meta embedding captures global context across modalities; (2) **robustness to missing modalities**, by dynamically adapting to incomplete inputs; and (3) **parameter efficiency**, as a single set of parameters that can generalize across all tasks.

After obtaining the integrated meta embeddings, we feed them into a vision transformer for relation modeling. To enhance its representational capacity, we introduce Dual Mixture-of-Experts (DMoE). Specifically, the T-MoE is dedicated to capturing complex and diverse spatio-temporal matching patterns, while the M-MoE focuses on multimodal knowledge fusion, effectively decoupling cross-modal dependencies and mitigating feature conflicts. This DMoE formulation provides two advantages. First, by explicitly separating temporal matching from multimodal feature fusion, the model avoids optimizing heterogeneous objectives within a single layer, thereby simplifying learning and reducing interference. Second, the sparsely activated nature of MoE ensures that the increased capacity only introduces marginal additional computation, while substantially improving the ability to capture diverse spatiotemporal and multimodal cues.

Extensive experiments validate the effectiveness of our *OneTrackerV2*. On RGB tracking benchmarks, previous RGB-only trackers can only perform RGB tracking; by contrast, *OneTrackerV2* can handle multimodal tracking and yet still outperforms those RGB-only baselines on RGB tracking tasks. For RGB+X tracking benchmarks, previous methods typically require task-specific architectures or a two-stage fine-tuning pipeline, while *OneTrackerV2* surpasses these approaches, only requiring a single step training and a unified architecture with shared parameters. Further analyses on robustness and model compression demonstrate that *OneTrackerV2* maintains high accuracy under missing modalities and compressed settings, highlighting its scalability, robustness, and efficiency.

In summary, the contributions can be summarized as follows: (1) We introduce *OneTrackerV2*, a unified framework that supports diverse multimodal tracking tasks through one-step training with a shared architecture and parameters, enabling scalability, robustness, and generalization across modalities. (2) We propose a novel Meta Merger module to aggregate RGB and multimodal features into a unified space, enabling efficient and effective cross-modal interaction while maintaining robustness under missing-modality conditions. (3) We propose DMoE, which decouples spatio-temporal relation modeling from multimodal feature embedding. This separation expands the representational space and improves overall tracking performance with negligible computational overhead. (4) Extensive experiments demonstrate that *OneTrackerV2* achieves state-of-the-art performance across five tracking tasks and twelve benchmarks. Moreover, *OneTrackerV2* maintains high efficiency and exhibits strong robustness under model compression and modality-missing scenarios.

## 2 RELATED WORKS

**RGB Tracking.** Visual object tracking aims to localize the target object in each video frame based on its initial appearance. In RGB tracking, only RGB images are used as input. Early ap-

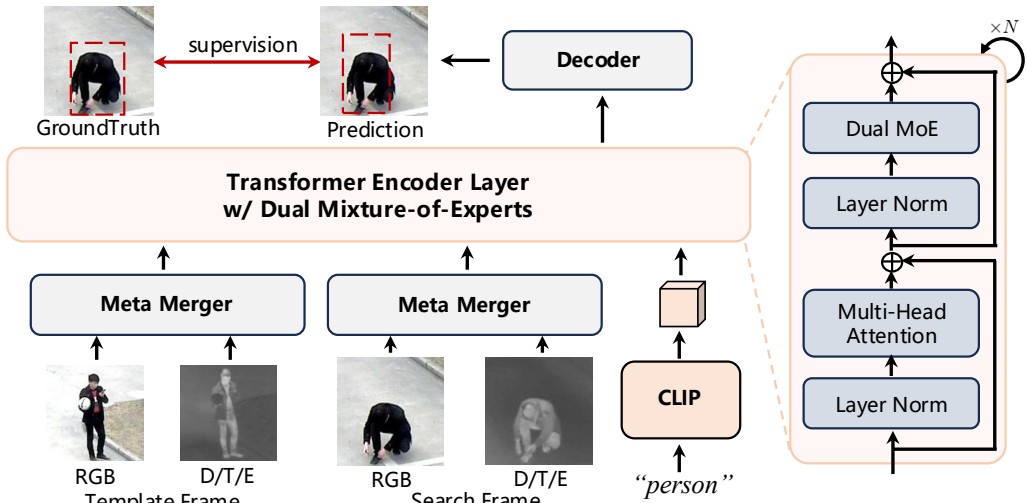

Figure 2: **Model Structure of *OneTrackerV2***. ***OneTrackerV2*** adopts a unified architecture that supports both RGB and RGB+X tracking tasks with shared parameters. RGB and multimodal inputs are first projected into a unified embedding space via Meta Merger, enabling effective cross-modal interaction. The fused features are then processed by a Vision Transformer with DMoE, which models rich spatio-temporal relations while decoupling multimodal dependencies for robust and efficient tracking.

proaches (Bertinetto et al., 2016; Bolme et al., 2010; Chen et al., 2021; Danelljan et al., 2019; Henriques et al., 2014; Li et al., 2019a; Yan et al., 2021a; Zhang et al., 2020) adopted a two-stream pipeline, where feature extraction and relation modeling were performed separately. Recently, one-stream pipelines (Bai et al., 2024; Chen et al., 2022; 2023b; Cui et al., 2022; 2023; Gao et al., 2023; Wei et al., 2023; Ye et al., 2022; Zhou et al., 2023a; 2024; Zheng et al., 2024; Liang et al., 2025) have become dominant, integrating these two stages into a unified process. These models are built upon Vision Transformers with stacked encoder layers. However, these approaches remain restricted to RGB-only inputs and are incapable of leveraging multimodal features. In contrast, our ***OneTrackerV2*** can process diverse multimodal inputs without relying on task-specific architectures.

**RGB+X Tracking.** In certain scenarios, relying solely on RGB images can be limiting. Thus, RGB+X tracking tasks are introduced, where additional modalities are incorporated to complement the weaknesses of RGB-based tracking. Depending on the input modality, RGB+X tracking can be categorized into four types: RGB+D (Depth) tracking (Yan et al., 2021b; Kristan et al., 2022), RGB+T (Thermal) tracking (Li et al., 2021; 2019b), RGB+E (Event) tracking (Wang et al., 2023), and RGB+N (Language) tracking (Li et al., 2017; Fan et al., 2019; Wang et al., 2021).

Existing RGB+X tracking methods primarily rely on separated architectures, where task-specific networks (Yan et al., 2021b; Li et al., 2021; Wang et al., 2023; 2021) are designed and trained independently for each modality combination. While effective, this strategy requires extensive architectural customization and separate training for every task. To alleviate this, subsequent works (Zhu et al., 2023; Hong et al., 2024b; Hou et al., 2024) adapt pretrained RGB trackers to downstream RGB+X tasks through fine-tuning, enabling cross-modality transfer. However, these methods depend on multi-stage training and often suffer from sub-optimal performance. SUTrack (Chen et al., 2025) attempts to handle multiple modalities within a single model, but it remains vulnerable to performance degradation under missing-modality scenarios. To address these limitations, we introduce ***OneTrackerV2***, a unified multimodal tracking framework that achieves robust performance without task-specific designs or multi-stage training.

**Mixture of Experts.** Mixture-of-experts (MoE) increases model capacity via conditional computation over multiple specialized experts and has been widely applied in large language models (Dai et al., 2024; Fedus et al., 2022; Yang et al., 2025). TIn visual object tracking, several studies have explored MoE (Tan et al., 2025; Cai et al., 2025; Tan et al., 2024; Guo et al., 2024), typically either using MoE to transfer pretrained RGB trackers to RGB+X settings or restricting training to RGB-only inputs. In contrast, we introduce Dual Mixture-of-Experts (DMoE) that combines a shared expert with a T-MoE and a M-MoE, explicitly decoupling temporal dynamics from multimodal in-

tegration. Equipped with DMoE, ***OneTrackerV2*** operates within a unified architecture with shared parameters and supports both RGB and RGB+X tracking with a one-step training.

# 3 ONETRACKER V2

## 3.1 OVERALL ARCHITECTURE

As shown in Figure 2, ***OneTrackerV2*** is designed as a unified framework for multimodal tracking, aiming to demonstrate how a single model can be trained to generalize across diverse modalities. ***OneTrackerV2*** takes as input the template and search frames, which may consist of RGB along with auxiliary modalities (*e.g.*, thermal, depth, event, or language). To bridge multimodal inputs, we first introduce Meta Merger, which projects RGB and multimodal features into a shared embedding space and enables flexible cross-modal interaction. This unified representation is then processed by a Vision Transformer backbone for relation modeling and target localization. To further enhance capacity without sacrificing efficiency, we introduce a Dual Mixture-of-Experts (DMoE), where the T-MoE branch focuses on spatio-temporal relation modeling, and the M-MoE branch addresses multimodal knowledge integration. Finally, the decoder (Ye et al., 2022; Chen et al., 2025) outputs bounding box predictions. Through this design, ***OneTrackerV2*** demonstrates that a tracker can be trained in a one-step with unified architecture and parameters, instead of relying on multi-stage adaptation or task-specific branches. This provides a general recipe for building scalable and robust multimodal trackers.

A central challenge in multimodal tracking is how to integrate heterogeneous modalities into a unified representation while remaining robust to missing modalities. Existing approaches often rely on task-specific branches (Hou et al., 2024), which substantially increase computational cost, or on naive concatenation with RGB features (Chen et al., 2025), which limits cross-modal interaction and yields suboptimal representations. To overcome these limitations, we introduce the Meta Merger, a lightweight yet effective module that unifies multimodal information into a shared embedding space and supports robust, modality-agnostic interaction.

## 3.2 META MERGER

A central challenge in multimodal tracking is how to integrate heterogeneous modalities into a unified representation while remaining robust to missing modalities. Existing approaches often rely on task-specific branches (Hou et al., 2024), which substantially increase computational cost, or on naive concatenation with RGB features (Chen et al., 2025), which limits cross-modal interaction and yields suboptimal representations. To overcome these limitations, we introduce the Meta Merger (as shown in Figure 3), a lightweight yet effective module that unifies multimodal information into a shared embedding space and supports robust, modality-agnostic interaction.

Concretely, given RGB frames and auxiliary modality frames (e.g., depth, thermal, event, or language), we first project them into feature maps using a shared patch embedding layer, ensuring consistency across modalities. For the RGB-only setting, the second branch reuses the RGB input to maintain structural uniformity. To enhance feature quality, we apply spatial and channel attentions on each modality branch, emphasizing informative cues before fusion.

Specifically, given the RGB frame and X modality frame, we first utilize a shared patch embedding layer and obtain corresponding feature maps $F_{rgb}$ and $F_x$. It is worth noting that for RGB tracking task, we use the same RGB images to replace the X images. Then, spatial and channel attentions are adopted to enhance both features, which can be illustrated as following:

$$W_{rgb}^{spatial} = \sigma(\text{Conv}(\text{AvgPool}(F_{rgb})) + \text{Conv}(\text{MaxPool}(F_{rgb}))),$$
$$W_{rgb}^{channel} = \sigma(\text{Linear}(\text{AvgPool}(F_{rgb})) + \text{Linear}(\text{MaxPool}(F_{rgb}))), \quad (1)$$
$$F_{rgb}^{'} = F_{rgb} \odot W_{rgb}^{spatial} \odot W_{rgb}^{channel} + F_{rgb},$$

where $\odot$ denotes element-wise multiplication, $W_{rgb}^{spatial}$ and $W_{rgb}^{channel}$ act as the spatial and channel attention weights respectively. For $F_x$, we also apply the same operations, which are omitted here.

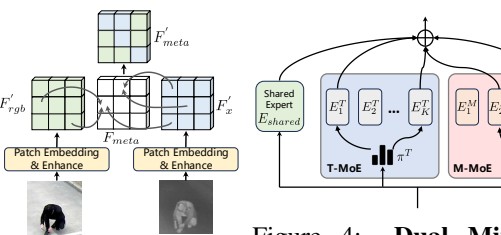

Figure 3: **Meta Merger.** Meta Merger unifies RGB and multimodal input into a shared space.

Figure 4: **Dual Mixture-of-Experts.** We introduce DMoE to decouple spatio-temporal relation modeling and multimodal feature integration.

Figure 5: **Details of *OneTrackerV2* variants**. We present four variants of *OneTrackerV2* with different input resolutions and model sizes. Meta Merger and DMoE introduce only marginal increases in parameters and computation. For clarity, the parameters and FLOPs of the text encoder are omitted here.

| Model | Search Resolution | Template Resolution | # Params (M) | FLOPs (G) | Speed (FPS) |
|---|---|---|---|---|---|
| *OneTrackerV2*-B224 | $224 \times 224$ | $112 \times 112$ | 80.2 | 23.8 | 72.4 |
| *OneTrackerV2*-B384 | $384 \times 384$ | $192 \times 192$ | 80.2 | 70.0 | 42.2 |
| *OneTrackerV2*-L224 | $224 \times 224$ | $112 \times 112$ | 271.1 | 77.7 | 46.6 |
| *OneTrackerV2*-L384 | $384 \times 384$ | $192 \times 192$ | 271.1 | 227.9 | 23.4 |

After that, a learnable meta embedding $F_{meta}$ is introduced as a global mediator to interact with both RGB and multimodal features:

$$F'_{meta} = \text{Conv}(\text{Conv}(F_{meta} + F'_{rgb}) + \text{Conv}(F_{meta} + F'_x) + F_{meta}). \quad (2)$$

Our Meta Merger leverages meta embedding as a central information hub. Instead of directly concatenating or stacking modality-specific features, the meta embedding interacts with both RGB and auxiliary features through lightweight convolutional transformations. This interaction allows the meta embedding to absorb, align, and redistribute cross-modal information, resulting in a compact and modality-agnostic semantic representation. Our Meta Merger features several advantages: (1) **Unified Modalities Bridge.** Meta embedding serves as a learnable bridge that harmonizes heterogeneous inputs, ensuring effective multimodal information alignment without hand-crafted fusion rules or modality-specific architectures.. (2) **Global Semantic Integration.** Meta embedding distills the most salient cues into a compact, globally consistent representation that benefits relation modeling downstream.. (3) **Robustness to Missing Modalities.** Since fusion is centered on meta embedding rather than direct concatenation, Meta Merger remains stable and effective even when some modalities are absent, which is unavaible in previous works (Chen et al., 2025). (4) **Lightweight.** Compared with approaches that use additional branches and double the computational cost (Hou et al., 2024), our Meta Merger introduces minimal overhead while enabling scalable multimodal learning.

### 3.3 DUAL MIXTURE-OF-EXPERTS

**Structure of DMoE.** After fusing multimodal features via Meta Merger, a Vision Transformer backbone is utilized for relation modeling. We introduce Dual Mixture-of-Experts (DMoE). The structure of DMoE is shown in Figure 4. Different from previous MoE designs (Fedus et al., 2022; Dai et al., 2024), which typically focus on a single objective, our DMoE explicitly decouples spatio-temporal relation modeling and multimodal feature integration within the same framework. DMoE consists of three kinds of experts: shared expert $E_{shared}$, T-MoE (Temporal-MoE), and M-MoE (Multimodal-MoE). For each token $x \in \mathbb{R}^d$, DMoE computes its output as:

$$y = E_{shared}(x) + \underbrace{\sum_{i \in S_k^T} \hat{g}_i^T(x) \cdot E_i^T(x)}_{y^T} + \underbrace{\sum_{i \in S_k^M} \hat{g}_i^M(x) \cdot E_i^M(x)}_{y^M}, \quad (3)$$

where $S_k^T = \text{top-}k(g_i^T(x))$ is the selected $k$ experts for T-MoE, $g_i^T(x) = \frac{\exp(\pi_i^T(x))}{\sum_{j=1}^{K} \exp(\pi_j^T(x))}$ denotes the gating weights after softmax, $\pi_i^T(\cdot)$ is origin gating value of T-MoE for the $i$-th expert, $\hat{g}_i^T(x) = \frac{g_i^T(x)}{\sum_{j \in S_k^T} g_j^T(x)}$ is the renormalized gating weights of $\text{top-}k$ experts, and $K$ is the number of experts. The gating process for M-MoE is defined analogously with $\pi_i^M(x)$, $g_i^M(x)$, and $\hat{g}_i^M(x)$. Each expert $E$ itself is implemented as a simple low-rank projection: inputs are mapped into a latent dimension $r$, transformed non-linearly, and then projected back. This design ensures high capacity without incurring prohibitive costs.

**Expert Decoupling.** If optimized jointly, T-MoE and M-MoE may collapse into learning overlapping patterns. To avoid redundancy and encourage complementary knowledge, we introduce a dissimilarity loss that penalizes cosine similarity between outputs of T-MoE ($y^T$) and M-MoE ($y^M$):

$$\mathcal{L}_{dis} = (\cos(y^T, y^M))^2 = (\frac{<y^T, y^M>}{||y^T||||y^M||})^2. \tag{4}$$

Intuitively, through $\mathcal{L}_{dis}$, we explicitly encourages T-MoE and M-MoE to remain decorrelated and enhance the diversity of learned features.

**Multimodal Router Cluster.** Although $\mathcal{L}_{dis}$ promotes diversity between T-MoE and M-MoE, the router of M-MoE remains disorganized across different modalities, failing to learn modality-specific expert assignments. To address this, we propose a router weights clustering regularization that aligns similar distributions to samples from the same modality and distinct distributions to samples from different modalities. Concretely, we compute similarity scores between routing logits $g^M(x)$ and encourage: (1) higher similarity for samples from the same modality; and (2) lower similarity for samples from different modalities. This is formalized as:

Specifically, given the router logits $g^M(x_i) \in \mathbb{R}^K$ for a sample $x_i$ in a batch of size B, we compute the similarity matrix among routing distributions as $S_{ij} = <g^M(x_i), g^M(x_j)>$. Then, we construct same-task and different-task masks based on the task indices. For samples from the same task, their similarity is encouraged to exceed a margin $\delta$ above the random baseline $(1/K)$, while for different tasks their similarity should be lower than a margin below $(1/K)$. Formally,

$$\mathcal{L}_{same} = \frac{1}{|M_{\text{same}}|} \sum_{(i,j) \in M_{\text{same}}} [max(0, (\frac{1}{K} + \delta) - S_{ij})],$$
$$\mathcal{L}_{diff} = \frac{1}{|M_{\text{diff}}|} \sum_{(i,j) \in M_{\text{diff}}} [max(0, S_{ij} - (\delta - \frac{1}{K}))], \tag{5}$$

where $M_{\text{same}}$ and $M_{\text{diff}}$ denote the index sets of sample pairs from the same and different tasks, respectively. The overall loss is then given by $\mathcal{L}_{cluster} = \mathcal{L}_{same} + \mathcal{L}_{diff}$. This router clustering enforces M-MoE router to produce consistent expert selections within the same modality while maintaining discriminative routing across modalities, thereby promoting both intra-modality coherence and inter-modality diversity. As a result, the model learns more discriminative and robust multimodal representations.

## 3.4 OPTIMIZATION OF ONETRACKERV2

**Loss Function.** Following pervious works (Ye et al., 2022; Chen et al., 2025), we adopt a similar loss formulation to supervise **OneTrackerV2**, and the overall objective is defined as:

$$\mathcal{L} = \mathcal{L}_{class} + \lambda_G\mathcal{L}_{IoU} + \lambda_{L_1}\mathcal{L}_{L_1} + \mathcal{L}_{task} + \lambda_{dis}\mathcal{L}_{dis} + \lambda_{cluster}\mathcal{L}_{cluster} + \lambda_{balance}\mathcal{L}_{balance}, \tag{6}$$

where $\mathcal{L}_{balance}$ is the balance loss, $\mathcal{L}_{class}$ and $\mathcal{L}_{task}$ is the same as that in (Chen et al., 2025), and $\lambda_G, \lambda_{L_1}, \lambda_{dis}, \lambda_{cluster}$, and $\lambda_{balance}$ are the hyperparameter with default values $\lambda_G = 2, \lambda_{L_1} = 5, \lambda_{dis} = 0.1, \lambda_{cluster} = 1$.

**Stochastic Modality Perturbation.** To enhance robustness and mitigate over-reliance on specific modalities, we introduce two stochastic training strategies. First, we apply modality replacement, where RGB and multimodal inputs are randomly swapped, encouraging model to learn modality-invariant representations. Second, inspired by (Tan et al., 2025), we also perform random modality masking, where either the RGB or multimodal input is randomly masked. Together, these strategies expose Meta Merger to diverse modality configurations, enabling it to learn more discriminative and robust embeddings, thereby improving overall model robustness.

Table 1: **Comparison with state-of-the-art trackers.** *OneTrackerV2* outperforms existing methods across 5 tasks and 12 benchmarks with a single training process, unified architecture, and shared parameters. We also compare whether each method supports multimodal inputs (MultiModal), employs a unified parameter (Unified Param), and is trained with a single step (Onestep Training).

| Method | Multi Modal | LaSOT AUC | $P_{Norm}$ | P | LaSOT$_{ext}$ AUC | $P_{Norm}$ | P | TrackingNet AUC | $P_{Norm}$ | P | GOT-10k AO | $SR_{0.5}$ | $SR_{0.75}$ | UAV123 AUC | NFS AUC |
|---|---|---|---|---|---|---|---|---|---|---|---|---|---|---|---|
| ***OneTrackerV2*-B224** | ✓ | 74.1 | 83.6 | 81.0 | 53.2 | 64.3 | 60.8 | 86.3 | 90.6 | 86.2 | 78.4 | 87.8 | 79.6 | 70.8 | 70.5 |
| ***OneTrackerV2*-B384** | ✓ | 75.4 | 85.4 | 83.8 | 54.1 | 65.1 | 61.3 | 87.2 | 91.1 | 87.6 | 79.6 | 89.4 | 81.3 | 71.1 | 70.9 |
| ***OneTrackerV2*-L224** | ✓ | 74.9 | 85.0 | 83.1 | 54.8 | 65.8 | 62.2 | 87.5 | 91.6 | 88.3 | 80.5 | 90.2 | 82.7 | 70.8 | 70.6 |
| ***OneTrackerV2*-L384** | ✓ | 76.1 | 86.0 | 84.4 | 55.2 | 66.1 | 62.9 | 88.6 | 92.5 | 89.0 | 81.3 | 91.8 | 83.9 | 71.0 | 70.8 |
| SUTrack-B224 (Chen et al., 2025) | ✓ | 73.2 | 83.4 | 80.5 | 53.1 | 64.2 | 60.5 | 85.7 | 90.3 | 85.1 | 77.9 | 87.5 | 78.5 | 70.9 | 69.8 |
| SUTrack-B384 (Chen et al., 2025) | ✓ | 74.4 | 83.9 | 81.9 | 52.9 | 63.6 | 60.1 | 86.5 | 90.7 | 86.8 | 79.3 | 88.0 | 80.0 | 70.4 | 69.3 |
| SUTrack-L224 (Chen et al., 2025) | ✓ | 73.5 | 83.3 | 80.9 | 54.0 | 65.3 | 61.7 | 86.5 | 90.9 | 86.7 | 81.0 | 90.4 | 82.4 | 70.9 | 69.8 |
| SUTrack-L384 (Chen et al., 2025) | ✓ | 75.2 | 84.9 | 83.2 | 53.6 | 64.2 | 60.5 | 87.7 | 91.7 | 88.7 | 81.5 | 89.5 | 83.3 | 70.4 | 69.3 |
| ARPTrack-B256 (Liang et al., 2025) | ✗ | 72.6 | 81.4 | 78.5 | 52.0 | 62.9 | 58.7 | 85.5 | 90.0 | 85.3 | 77.7 | 87.3 | 74.3 | 71.7 | 67.4 |
| SPMTrack-B384 (Cai et al., 2025) | ✗ | 74.9 | 84.0 | 81.7 | - | - | - | 86.1 | 90.2 | 85.6 | 76.5 | 85.9 | 76.3 | 71.7 | 67.4 |
| ODTrack-B384 (Zheng et al., 2024) | ✗ | 73.2 | 83.2 | 80.6 | 52.4 | 63.9 | 60.1 | 85.1 | 90.1 | 84.9 | 77.0 | 87.9 | 75.1 | - | - |
| LoRAT-B378 (Lin et al., 2024) | ✗ | 72.9 | 81.9 | 79.1 | 53.1 | 64.8 | 60.6 | 84.2 | 88.4 | 83.0 | 73.7 | 82.6 | 72.9 | 71.9 | 66.6 |
| ARTrackV2-256 (Bai et al., 2024) | ✗ | 71.6 | 80.2 | 77.2 | 50.8 | 61.9 | 57.7 | 84.9 | 89.3 | 84.5 | 75.9 | 85.4 | 72.7 | 69.9 | 67.6 |
| OneTracker-384 (Hong et al., 2024b) | ✓ | 70.5 | 79.9 | 76.5 | - | - | - | 83.7 | 88.4 | 82.7 | - | - | - | - | - |
| ARPTrack-L384 (Liang et al., 2025) | ✗ | 74.2 | 83.4 | 81.7 | 54.2 | 64.4 | 61.2 | 86.6 | 91.1 | 87.4 | 81.5 | 90.6 | 80.5 | 71.7 | 67.4 |
| SPMTrack-L384 (Cai et al., 2025) | ✗ | 76.8 | 85.9 | 84.0 | - | - | - | 86.9 | 91.0 | 87.2 | 80.0 | 89.4 | 79.9 | - | - |
| LoRAT-L378 (Lin et al., 2024) | ✗ | 75.1 | 84.1 | 82.0 | 56.6 | 69.0 | 65.1 | 85.6 | 89.7 | 85.4 | 77.5 | 86.2 | 78.1 | 72.5 | 66.7 |
| ODTrack-L384 (Zheng et al., 2024) | ✗ | 74.0 | 84.2 | 82.3 | 53.9 | 65.4 | 61.7 | 86.1 | 91.0 | 86.7 | 78.2 | 87.2 | 77.3 | - | - |
| ARTrackV2-L384 (Bai et al., 2024) | ✗ | 73.6 | 82.8 | 81.1 | 53.4 | 63.7 | 60.2 | 86.1 | 90.4 | 86.2 | 79.5 | 87.8 | 79.6 | 71.7 | 68.4 |

| Method | Unified Param | Onestep Training | DepthTrack F-Score | Re | Pr | LasHeR AUC | P | RGBT234 MSR | MPR | VisEvent AUC | P | TNL2K AUC | P | OTB99 AUC | P |
|---|---|---|---|---|---|---|---|---|---|---|---|---|---|---|---|
| ***OneTrackerV2*-B224** | ✓ | ✓ | 65.9 | 66.7 | 65.8 | 60.6 | 75.5 | 69.5 | 91.3 | 63.0 | 79.8 | 65.8 | 70.2 | 72.5 | 94.8 |
| ***OneTrackerV2*-B384** | ✓ | ✓ | 66.8 | 67.2 | 66.9 | 61.3 | 75.7 | 70.2 | 91.6 | 63.9 | 80.5 | 66.4 | 71.9 | 72.8 | 95.0 |
| ***OneTrackerV2*-L224** | ✓ | ✓ | 66.6 | 67.7 | 66.4 | 62.5 | 77.7 | 72.0 | 94.5 | 64.7 | 81.9 | 67.5 | 73.0 | 72.9 | 94.6 |
| ***OneTrackerV2*-L384** | ✓ | ✓ | 67.5 | 68.4 | 67.2 | 63.7 | 79.4 | 72.5 | 94.5 | 65.9 | 83.5 | 69.5 | 76.0 | 73.2 | 95.3 |
| SUTrack-B224 (Chen et al., 2025) | ✓ | ✓ | 65.1 | 65.7 | 64.5 | 59.9 | 74.5 | 69.5 | 92.2 | 62.7 | 79.9 | 65.0 | 67.9 | 70.8 | 93.4 |
| SUTrack-B384 (Chen et al., 2025) | ✓ | ✓ | 64.4 | 64.2 | 64.6 | 60.9 | 75.8 | 69.2 | 92.1 | 63.4 | 79.8 | 65.6 | 69.3 | 69.7 | 91.2 |
| SUTrack-L224 (Chen et al., 2025) | ✓ | ✓ | 64.3 | 64.6 | 64.6 | 61.9 | 77.0 | 70.8 | 94.6 | 64.0 | 80.9 | 66.7 | 70.3 | 72.7 | 94.4 |
| SUTrack-L384 (Chen et al., 2025) | ✓ | ✓ | 66.4 | 66.4 | 66.5 | 61.9 | 76.9 | 70.3 | 93.7 | 63.8 | 80.5 | 67.9 | 72.1 | 71.2 | 93.1 |
| CSTrack-L256 (Feng et al., 2025) | ✓ | ✗ | 65.8 | 66.4 | 65.2 | 60.8 | 75.6 | 70.9 | 94.0 | 65.2 | 82.4 | - | - | - | - |
| STTrack-B256 (Hu et al., 2025) | ✓ | ✗ | 63.3 | 63.4 | 63.2 | 60.3 | 76.0 | 66.7 | 89.8 | 61.9 | 78.6 | - | - | - | - |
| SeqTrackv2-L384 (Chen et al., 2023a) | ✓ | ✗ | 62.3 | 62.6 | 62.5 | 61.0 | 76.7 | 68.0 | 91.3 | 63.4 | 80.0 | 62.4 | 66.1 | 71.4 | 93.6 |
| SDSTrack-B256 (Hou et al., 2024) | ✗ | ✗ | 61.4 | 60.9 | 61.9 | 53.1 | 66.5 | 62.5 | 84.8 | 59.7 | 76.7 | - | - | - | - |
| UnTrack-B256 (Wu et al., 2024) | ✓ | ✗ | 61.0 | 60.8 | 61.1 | 51.3 | 64.6 | 62.5 | 84.2 | 58.9 | 75.5 | - | - | - | - |
| OneTracker (Hong et al., 2024b) | ✗ | ✗ | 60.9 | 60.4 | 60.7 | 53.8 | 67.2 | 64.2 | 85.7 | 60.8 | 76.7 | 58.0 | 59.1 | 69.7 | 91.5 |
| ViPT (Zhu et al., 2023) | ✗ | ✗ | 59.4 | 59.6 | 59.2 | 52.5 | 65.1 | 61.7 | 83.5 | 59.2 | 75.8 | - | - | - | - |

## 4 EXPERIMENTS

### 4.1 IMPLEMENTATION DETAILS

**Model Structure.** We develop a series of models trained with different model sizes and input resolutions to provide four variants of *OneTrackerV2*. All variants adopt HiViT (Zhang et al., 2022) initialized with Fast-iTPN (Tian et al., 2024) as the encoder. Number of experts $k$ and rank $r$ are set as 2 and 16. Detailed statistics of each model, including the number of parameters, FLOPs, and inference speed, are presented in Table 5. In addition, we employ CLIP-L (Radford et al., 2021) as the text encoder to extract language features. For tasks without text input, the parameters of the text encoder can be omitted.

**Training and Inference.** *OneTrackerV2* is trained on the combination of RGB and RGB+X tracking datasets, including LaSOT (Fan et al., 2019), TrackingNet (Muller et al., 2018), GOT-10k (Huang et al., 2019), COCO (Lin et al., 2014), VASTTrack (Peng et al., 2024), DepthTrack (Yan et al., 2021b), VisEvent (Wang et al., 2023), LasHeR (Li et al., 2021), and TNL2K (Wang et al., 2021). *OneTrackerV2* is optimized using AdamW (Loshchilov & Hutter, 2017) with a total of 300 training epochs, each consisting of 100,000 sampled image pairs. During training, we sample and mix data across these datasets. During inference, we apply a Hanning window penalty, following previous works (Ye et al., 2022). The template is updated every 25 frames, conditioned on an update confidence threshold of 0.7.

### 4.2 COMPARISONS WITH THE STATE-OF-THE-ART

**Main Results.** We evaluate *OneTrackerV2* against state-of-the-art RGB and RGB+X trackers across 5 tasks 12 benchmarks in Table 1. OneTrackerV2 consistently outperforms existing methods in terms of accuracy across all datasets. Specially, on RGB tracking benchmarks, *OneTrackerV2* surpasses prior RGB-only models, demonstrating that the unified design does not compromise single-modality performance. for RGB+X tracking, *OneTrackerV2* requires only a single training

Table 2: **Comparison on missing-modality benchmarks.** We evaluate *OneTrackerV2* and existing RGB+X trackers under scenarios where certain modalities are absent. Thanks to the proposed Meta Merger, *OneTrackerV2* consistently outperforms prior methods, demonstrating superior robustness to modality missing in multimodal tracking.

| Method | Unified Param | Onestep Training | DepthTrack$_{miss}$ | | | LasHeR$_{miss}$ | | RGBT234$_{miss}$ | | VisEvent$_{miss}$ | |
|---|---|---|---|---|---|---|---|---|---|---|---|
| | | | F-score | Re | Pr | AUC | P | MSR | MPR | AUC | P |
| ***OneTrackerV2*-B224** | ✓ | ✓ | 56.9 | 55.4 | 58.6 | 52.7 | 66.2 | 63.5 | 85.5 | 54.3 | 71.5 |
| STTrack (Hu et al., 2025) | ✗ | ✗ | 49.9 | 48.8 | 51.0 | 44.9 | 54.5 | 54.2 | 73.8 | 49.7 | 65.5 |
| SUTrack Chen et al. (2025) | ✓ | ✓ | 49.5 | 47.3 | 51.9 | 47.6 | 58.3 | 60.8 | 82.0 | 50.5 | 66.6 |
| SeqTrackv2 Chen et al. (2023a) | ✗ | ✗ | 45.0 | 40.9 | 50.0 | 39.9 | 50.0 | 49.9 | 70.8 | 43.1 | 57.6 |
| SDSTrack Hou et al. (2024) | ✗ | ✗ | 46.7 | 42.0 | 52.7 | 43.1 | 52.5 | 48.8 | 67.0 | 46.9 | 62.6 |
| ViPT Zhu et al. (2023) | ✗ | ✗ | 44.4 | 40.5 | 46.6 | 34.0 | 40.1 | 39.4 | 52.4 | 43.2 | 57.2 |
| MCITrack Kang et al. (2025) | ✗ | ✗ | 49.7 | 42.9 | 59.1 | 40.0 | 34.2 | 40.9 | 53.6 | 36.5 | 49.9 |

Table 3: **Ablation study.** We investigate the impact of each module on *OneTrackerV2*. We train the model with HiViT-B as backbone only for 180 epochs.

| # | Method | FPS ↑ | Params (M) | FLOPs (G) | LaSOT | TNL2K | LasHeR | DepthTrack | VisEvent | Average | DepthTrack$_{miss}$ | LasHeR$_{miss}$ |
|---|---|---|---|---|---|---|---|---|---|---|---|---|
| 1 | Baseline | 95 | 70.0 | 23.0 | 69.2 | 61.3 | 50.4 | 51.7 | 53.5 | 57.2 | 48.4 | 43.7 |
| 2 | + Meta Merger | 90(−5.3%) | 70.6(+0.9%) | 23.1(+0.4%) | 71.3 | 62.5 | 55.7 | 62.1 | 58.8 | 62.1 | 51.8 | 47.9 |
| 3 | + Stochastic Perturbation | 90(−5.3%) | 70.6(+0.9%) | 23.1(+0.4%) | 71.0 | 62.4 | 55.6 | 62.3 | 58.5 | 62.0 | 53.2 | 48.6 |
| 4 | + Single Mixture-of-Experts | 83(−11.6%) | 75.4(+7.7%) | 23.5(+2.2%) | 71.4 | 62.9 | 56.2 | 62.9 | 58.4 | 62.4 | 53.6 | 49.3 |
| 5 | + Dual Mixture-of-Experts | 72(−24.2%) | 80.2(+14.6%) | 23.8(+3.5%) | 71.6 | 63.3 | 56.8 | 63.2 | 59.2 | 62.8 | 54.1 | 49.8 |
| 6 | + Expert Decoupling | 72(−24.2%) | 80.2(+14.6%) | 23.8(+3.5%) | 71.6 | 63.3 | 56.8 | 63.2 | 59.2 | 62.8 | 54.7 | 50.2 |
| 7 | + Router Cluster | 72(−24.2%) | 80.2(+14.6%) | 23.8(+3.5%) | 72.0 | 63.7 | 57.5 | 64.2 | 60.2 | 63.5 | 54.7 | 50.2 |

process and a unified parameters, yet still outperforms existing approaches that rely on task-specific architectures or downstream fine-tuning. Notably, on TNL2K, *OneTrackerV2* achieves 69.5 AUC. These results clearly demonstrate the superiority of *OneTrackerV2*. By unifying training, architecture, and parameters across diverse RGB and RGB+X tasks, *OneTrackerV2* not only achieves state-of-the-art accuracy but also delivers strong robustness and scalability.

**Robustness Against Missing Modality.** Following FlexTrack (Bai et al., 2024), we conduct experiments on multiple RGB+X benchmarks under various missing-modality settings. As shown in Table 2, *OneTrackerV2* achieves significant improvements over previous methods. These results highlight not only the robustness of *OneTrackerV2* to modality absence, but also the effectiveness of the proposed Meta Merger in enabling stable and reliable multimodal fusion.

## 4.3    ABLATION STUDY

We conduct ablation studies on HiViT-B (Zhang et al., 2022) to examine how the modules we propose contribute to building a unified multimodal tracker. Note that in all ablation experiments, the models are trained for only 180 epochs.

**Meta Merger.** To demonstrate the effectiveness of the proposed Meta Merger module, we conduct ablation experiments, and the results are shown in row #2 and #3 of Table 3. Compared with the baseline, integrating the Meta Merger yields more effective fusion of multi-modal features, leading to consistent performance gains. Besides, the computational cost of Meta Merger is negligible (only additional 0.4% FLOPs and 0.9% parameters).

Moreover, under modality-missing scenarios, the combination of Meta Merger and Stochastic Perturbation significantly enhances the robustness of the model. This indicates that our design not only improves the overall representation ability in multi-modal fusion, but also equips the model with stronger and robust adaptability to incomplete or noisy modality inputs.

**Dual Mix-of-Experts.** We further investigate the impact of the proposed Dual MoE design, with results summarized in Table 3. When we simply increase the capacity of a single MoE (row #4), model exhibits only moderate performance improvement. Extending this design to a straightforward Dual MoE brings only marginal additional gains, indicating that naively stacking MoEs is insufficient to fully exploit their potential. Then, we incorporate two key enhancements: Expert Decoupling and Router Clustering. With these two modifications, Dual MoE achieves significant performance gains, demonstrating the effectiveness and necessity of our Dual MoE. Moreover, the introduction of Dual MoE only incurs an interpolation of 14.6% parameters and 3.5% FLOPs, which is a relatively small price to pay given the significant performance gains achieved.

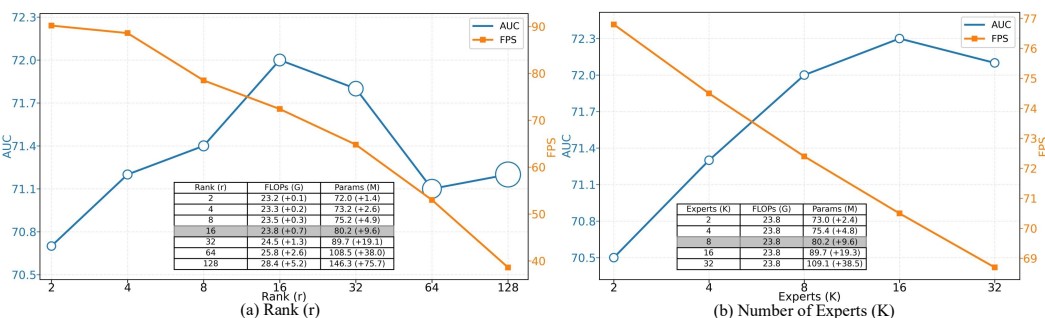

Figure 6: **Analysis of expert hyperparameters.** We show the impact of (a) different rank ($r$) and (b) different number of experts ($K$) on model parameters, computational cost, FPS, and accuracy.

Table 4: **Compression Performance.** We compress *OneTrackerV2-B224* into a 6-layer variant (*OneTrackerV2*-B224-Compress) following CompressTracker (Hong et al., 2024a). *OneTrackerV2*-B224-Compress achieves the balance between accuracy and efficiency.

| Method | LaSOT | | | DepthTrack | | | LasHeR | | VisEvent | | TNL2K | | FPS |
|---|---|---|---|---|---|---|---|---|---|---|---|---|---|
| | AUC | $P_{norm}$ | P | F | Re | Pr | AUC | P | AUC | P | AUC | P | |
| *OneTrackerV2*-B224-Compress | 73.0 | 83.1 | 81.1 | 64.6 | 65.0 | 65.2 | 59.7 | 74.9 | 62.0 | 79.2 | 64.7 | 68.5 | 159 |
| SUTrack-T224 (Chen et al., 2025) | 69.6 | 79.3 | 75.4 | 61.7 | 62.1 | 61.2 | 53.9 | 66.7 | 58.8 | 75.7 | 60.9 | 62.3 | 100 |
| STTrack-B256 (Hu et al., 2025) | - | - | - | 63.3 | 63.4 | 63.2 | 60.3 | 76.0 | 61.9 | 78.6 | - | - | 36 |
| OneTracker-384 (Hong et al., 2024b) | 70.5 | 79.9 | 76.5 | 60.9 | 60.4 | 60.7 | 53.8 | 67.2 | 60.8 | 76.7 | 58.0 | 59.1 | - |
| SeqTrackerV2-B224 (Chen et al., 2023b) | 69.9 | 79.7 | 76.3 | 63.2 | 63.4 | 62.9 | 55.8 | 70.4 | 61.2 | 78.2 | 57.5 | 59.7 | 40 |

**Hyper Parameters of Experts.** We further conduct a detailed study on the impact of hyperparameters in DMoE, with a particular focus on the rank of expert projection ($r$) and the number of experts ($K$). As illustrated in Figure 6, increasing the rank consistently improves model performance in the beginning, which indicates that higher-rank projections enhance the expressive capacity of the Mixture-of-Experts and allow the model to capture richer patterns. However, when the rank exceeds 16, the performance starts to drop slightly. This phenomenon suggests that overly large ranks may introduce redundancy into the learned representations, which limits further improvement. Moreover, as the rank increases, inference speed inevitably decreases, clearly highlighting a trade-off between efficiency and accuracy. Figure 6 (a) demonstrates that setting the rank to 16 achieves the optimal balance between inference efficiency and tracking accuracy. Similarly, Figure 6 (b) examines the effect of the number of experts. Enlarging the expert count generally leads to better performance, since additional experts expand the representational space and enable the model to learn more diverse knowledge. Nevertheless, excessively increasing the number of experts also results in prohibitively high parameter overhead and computational burden, which is impractical in real applications. Taking both accuracy and efficiency into account, we therefore adopt 8 experts as the optimal configuration for our *OneTrackerV2*, striking a desirable balance between representational capacity and model complexity.

**Compression Effectiveness.** Following CompressTracker (Hong et al., 2024a), we compress our *OneTracker*-B224 into a lightweight variant *OneTracker*-B224-Compress with only six layers. As shown in Table 4, the compressed model, *OneTracker*-B224-Compress, achieves a $2.2\times$ speedup while incurring only about a $2\%$ performance drop on both RGB and RGB+X tracking benchmarks. Specifically, *OneTrackerV2*-B224-Compress reaches 159 FPS, surpassing SuTrack-T224 (Chen et al., 2025) in both speed and accuracy. For instance, *OneTrackerV2*-B224-Compress outperforms SuTrack-T224 by 3.4 AUC on LaSOT (73.0 *vs* 69.6) and by 5.8 AUC on LasHeR (59.7 *vs* 53.9). This result demonstrates the effectiveness of *OneTracker* under model compression: even in a compact form, *OneTrackerV2* preserves strong multimodal fusion and tracking capability.

## 5 CONCLUSION

In this work, we present *OneTrackerV2*, a unified paradigm for multimodal tracking. Instead of designing modality-specific structure or relying on multi-stage fine-tuning, we demonstrate that a single framework with Meta Merger for modality unification and DMoE for decoupled expert learning can achieve robust multimodal tracking with one-step training. Extensive experiments show that *OneTrackerV2* not only surpasses existing models in both RGB and RGB+X tracking tasks, but also remains robust under missing modalities and effective in compressed settings.

# 6 ETHICS STATEMENT

This work focuses on advancing multimodal visual object tracking through the design of a unified and efficient framework. All datasets used in our experiments are publicly available and widely adopted in the tracking community. We strictly follow the terms of use of each dataset and ensure that no private or sensitive data are involved. Our method is intended solely for academic research and applications such as robotics, autonomous systems, and video analysis. Nevertheless, we acknowledge that tracking technologies may potentially be misused for surveillance or privacy-invasive purposes. We therefore encourage responsible use of our model and stress that any deployment should comply with local laws, ethical guidelines, and privacy protection standards.

# 7 REPRODUCIBILITY STATEMENT

To ensure reproducibility, we provide detailed descriptions of our model architectures, training settings, and hyperparameters in main paper. All datasets used are publicly available, and their usage are explicitly referenced in the paper.

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

# A APPENDIX

## A.1 THE USE OF LARGE LANGUAGE MODELS (LLMS)

In this work, Large Language Models (LLMs) are used solely as general-purpose assistive tools to help polish and improve the clarity of the writing. Authors take full responsibility for all content in the paper, including text that was refined using LLMs, and confirm that no part of the manuscript generated by LLMs constitutes plagiarism or scientific misconduct.

## A.2 LIMITATION

While *OneTrackerV2* demonstrates strong performance across diverse RGB and RGB+X tracking tasks, there are several limitations.Training a unified multimodal tracker still requires access to multiple datasets with sufficient coverage of different modalities, which may not always be available. Performance of *OneTrackerV2* may degrade when encountering previously unseen or entirely new modalities that were not present during training.

