# OpenReview forum: "OneTrackerV2: Unified Multimodal Visual Object Tracking with Mixture of Experts"
_ICLR.cc/2026/Conference — ICLR 2026 Conference Withdrawn Submission_

### Official Review · Reviewer_Yfko · 2025-11-01

**Soundness:** 3
**Presentation:** 3
**Contribution:** 1
**Rating:** 4
**Confidence:** 4

**Summary:**

This paper attempts to address the well-known challenge of creating a unified visual object tracker , aiming to replace the common paradigms of using task-specific models or multi-stage adaptation. The authors' goal is to use a single, end-to-end trained model with shared parameters to handle both standard RGB and various RGB+X (e.g., Depth, Thermal, Language) tracking tasks.

**Strengths:**

* **Ambitious Problem Formulation:** The paper tackles the significant challenge of unifying diverse tracking tasks (RGB and RGB+X) into a single, efficient, end-to-end trained model. This represents a clear effort to move beyond the less scalable "one model per task" or multi-stage adaptation paradigms.
* **Focus on Modality Robustness:** A key strength is the explicit focus on handling missing or corrupted modalities. The architecture (Meta Merger) and training strategy (Stochastic Perturbation) are specifically designed to address this practical and challenging scenario, rather than assuming complete, high-quality inputs.
* **Systematic Ablation Study:** The paper provides a comprehensive ablation study that systematically evaluates the contribution of each proposed component. This includes isolating the effects of the Meta Merger, the DMOE, and the specific auxiliary losses ($\mathcal{L}_{dis}$, $\mathcal{L}_{cluster}$), which helps to justify the final architectural design.

**Weaknesses:**

* **Limited Conceptual Novelty:** The paper's primary weakness is its incremental novelty. It presents a strong engineering effort in integrating existing ideas, but the core concepts are not new. Unified models, the application of MoE to vision, and the use of learnable tokens for fusion are all pre-existing. The DMOE's specific split of experts (temporal vs. modal) is an architectural choice, not a new paradigm.
* **Overstated Contribution of Meta Merger:** The "Meta Merger" is framed as a key innovation. However, its design—using a learnable embedding ($F_{meta}$) as a central hub for convolutional fusion—is a variant of common "query-based" or "token-based" fusion patterns found in many multimodal architectures. The "Meta" branding seems to overstate the conceptual uniqueness of this component.
* **Questionable Semantic Decoupling in DMOE:** The paper claims DMOE achieves a *semantic* decoupling of spatio-temporal (T-MoE) and multimodal (M-MoE) knowledge, enforced by a simple dissimilarity loss on the outputs. This claim is questionable, as these two concepts are often deeply entangled. For example, the utility of thermal data (modal) is directly tied to its spatio-temporal properties (like a heat signature). It is unclear if the loss achieves true semantic separation (e.g., T-MoE learns *generic motion*, M-MoE learns *specific modality features*) or if it merely forces two expert groups to learn decorrelated functions that do not map cleanly to the claimed roles.

**Questions:**

* **Qualitative Analysis of DMOE Decoupling:** Could the authors provide a more qualitative analysis to support the claim that DMOE achieves a *semantic* decoupling of spatio-temporal versus modal knowledge? For example, could expert activation maps be visualized for T-MoE and M-MoE on a task like RGB+T tracking? One would expect T-MoE to activate on generic motion cues, while M-MoE activates on areas where the thermal modality provides unique information. This would be more convincing than just showing that the $\mathcal{L}_{dis}$ loss improves performance metrics.

* **Justification for $\mathcal{L}_{cluster}$:** The $\mathcal{L}_{cluster}$ loss encourages M-MoE experts to become *modality-specific*. This seems counter-intuitive for a *unified* model, where one might desire experts to learn abstract, *cross-modal* functions (e.g., an expert for "low-light enhancement" that could leverage *either* Thermal or Event data). Why is forcing modality-specialization preferable to allowing the experts to learn more abstract, modality-agnostic roles?

* **Meta Merger Baseline Definition:** The ablation study in Table 3 compares the "Meta Merger" (Row 2) against an undefined "Baseline" (Row 1). Could the authors explicitly define what this baseline model entails? To better situate the novelty of Meta Merger, how does it compare against other well-established fusion mechanisms, such as simple concatenation or a standard cross-attention module, when plugged into the same backbone?

---

### Official Review · Reviewer_4Zcd · 2025-11-01

**Soundness:** 3
**Presentation:** 3
**Contribution:** 2
**Rating:** 6
**Confidence:** 4

**Summary:**

The paper proposes OneTrackerV2, a single unified multimodal visual object tracker that can handle RGB and RGB+X within one model. The core idea is to use a Meta Merger to project heterogeneous modalities into a shared space and a DMoE to decouple temporal/spatial modeling from multimodal fusion.

**Strengths:**

Overall, the paper is technically coherent and empirically well supported:

- a) Clear motivation for a single tracker across RGB and RGB+X scenarios. The paper articulates well why having separate RGB, RGB+X, … trackers is costly, and why adaptation-based methods  still rely on task-dependent weights. The “one-step, one-arch, one-parameter” goal is explicit and consistent throughout.

- b). Method is technically coherent:  The two core modules(e.g. Meta Merger + DMoE ) are conceptually simple but map nicely to the stated challenges.

- c). The compressed variant shows that the learned representation is not overly brittle and can be deployed in lighter settings.

**Weaknesses:**

Overall, the key ideas are solid but parts of the paper feel more like a strong integration of known components than a fundamentally new tracking paradigm:

- a). Both the meta-token style fusion and the use of MoE for heterogeneous inputs are established ideas; the paper would benefit from a sharper argument on why a single MoE or a simpler gated fusion is insufficient in this unified tracking regime.

- b). The “one-step large mixed training” is central to the contribution, but the paper currently does not specify sampling ratios, modality-drop schedules, or balancing rules across very imbalanced datasets, which weakens reproducibility and fairness.

**Questions:**

- a). 	The dual-branch MoE introduces non-trivial computation and latency; for a tracking task this should be discussed more explicitly (e.g., whether sparse routing or single-branch inference is possible).

- b). Overall, the method would be much stronger if the training recipe and expert behavior were made more explicit.

---

### Official Review · Reviewer_2h1R · 2025-11-01

**Soundness:** 2
**Presentation:** 2
**Contribution:** 2
**Rating:** 4
**Confidence:** 5

**Summary:**

This paper proposes OneTrackerV2, a unified multi-modal visual object tracking framework that handles multiple input modalities within a single model. OneTrackerV2 uses one architecture and one-step end-to-end training to support diverse tracking tasks. The core methodological contributions are two novel modules: Meta Merger, which learns to embed and fuse RGB and auxiliary modality features into a shared representation space for effective cross-modal interaction, and Dual Mixture-of-Experts (DMoE), which consists of a spatio-temporal expert (T-MoE) and a multimodal expert (M-MoE). The authors also introduce a dissimilarity loss to encourage the two expert branches to learn diverse representations, and use stochastic modality perturbation during training to improve robustness to missing modalities. Empirically, OneTrackerV2 is shown to achieve state-of-the-art performance across 5 different tracking tasks, outperforming specialized trackers on each individual task.

**Strengths:**

1. The paper tackles the challenge of training a single model for multiple tracking modalities, addressing known issues of prior works. This unified approach is appealing for scalability and maintenance, and the authors clearly articulate why a one-model-for-all tracker is beneficial.

2. The proposed modules serve distinct architectural roles. Meta Merger employs a learnable fusion mechanism that maps modality-specific features into a shared embedding. This replaces direct concatenation and preserves feature consistency under missing modalities. Dual Mixture-of-Experts (DMoE) extends the MoE framework to tracking by dividing experts into spatio-temporal and multi-modal branches, separating motion modeling from sensor fusion.

3. One of the paper’s biggest strengths is the comprehensive experimental validation. OneTrackerV2 is evaluated on a broad set of benchmarks spanning five different tracking scenarios. Impressively, the unified model achieves state-of-the-art results across most of these benchmarks.

**Weaknesses:**

1. Figure 1 aims to provide a high-level comparison between OneTracker V2 and prior models, but the comparison is conceptually flawed. The figure contrasts OneTracker V2 primarily against OneTracker, overlooking SUTrack, which already incorporates all the claimed advantages—one-step training, unified parameters, unified architecture, and robustness to missing modalities.

2. Although the paper presents a unified framework, most components are adaptations or recombinations of established techniques. The notion of unifying multimodal tracking has been studied previously, and this work constitutes an incremental refinement in training and architecture. The Mixture-of-Experts concept, adapted from NLP and large-scale models, is applied here in a new context but does not represent a fundamentally new paradigm. Overall, the methodological originality is moderate, relying on existing transformer and MoE principles rather than introducing new theoretical developments.

3. The work lacks sufficient analysis to explain why the proposed components function as intended. The DMoE’s separation of temporal and multimodal processing yields empirical gains but is not examined to reveal what each expert learns or why the division is optimal. The claim that M-MoE mitigates cross-modal feature conflicts is unsubstantiated beyond accuracy improvements. Similarly, the Meta Merger is presented as superior to concatenation without analysis of its internal behavior under missing modalities. The absence of qualitative or quantitative evidence leaves these mechanisms as black-box enhancements, limiting interpretability and theoretical understanding of the model’s effectiveness.

4. The framework introduces multiple elements at once (fusion module, dual-expert transformer, extra loss, custom augmentations), raising system complexity. Component interactions are hard to parse, hampering reproducibility and clean attribution of gains; even the ablation bundles perturbations, obscuring individual effects.

5. Key baselines are absent—e.g., a unified multi-task model with simple concatenation (no Meta Merger/MoE) or a “unified w/o Meta Merger & DMoE” control—to isolate module contributions vs. multi-tasking. Training logistics (dataset mixing, task balance, scheduling) are unspecified. Compression results lack method and trade-off details (size vs. accuracy). These omissions are not fatal but leave questions that weaken the completeness of the evaluation.

**Questions:**

See weaknesses.

---

### Official Review · Reviewer_pKzm · 2025-11-04

**Soundness:** 3
**Presentation:** 3
**Contribution:** 3
**Rating:** 6
**Confidence:** 5

**Summary:**

Summary:
OneTrackerV2 is a single tracker that works across RGB videos and RGB plus extra modalities (depth, thermal, event, or language). It introduces a Meta Merger that puts all modalities into one shared feature space, which makes the model more stable when some inputs are missing. It also adds a Dual Mixture-of-Experts (DMoE): one set of experts focuses on motion and temporal cues, while another handles cross-modal fusion, with losses that keep them complementary rather than redundant. Trained end-to-end once with shared parameters, OneTrackerV2 delivers state-of-the-art results on five tracking settings and twelve benchmarks, and stays strong under missing-modality and compressed (faster) variants.

**Strengths:**

1. The proposed Meta Merger provides a lightweight multimodal fusion that scales and remains robust under missing inputs.
2. This paper proposes the Dual MoE which separates temporal modeling from modality fusion and the unified, one-pass training removes task-specific heads and multi-stage fine-tuning, make it easy to use.
3. The evaluation across RGB and RGB+X (D/T/E/N) shows consistent gains over SOTA baselines (e.g., SUTrack, OneTracker, SDSTrack).

**Weaknesses:**

1. Using a shared meta-embedding to fuse all modalities risks letting the dominant RGB stream shape the unified space when training data are imbalanced. Please (i) report the per-modality data distribution and any re-sampling/re-weighting you used, (ii) detail the stochastic modality perturbation and show performance with masking rate in modality proportions to quantify sensitivity.

2. The theoretical explanation for the Dual MoE design is insufficient. The distinction between T-MoE (temporal) and M-MoE (modality) is not clearly reflected in the architecture or learning objectives, making the split appear largely nominal. The authors should clarify how temporal and modality-specific features are explicitly separated in practice, describe the mechanisms that ensure true decoupling (e.g., input routing, expert assignments, or loss constraints), and provide empirical or theoretical evidence that this separation contributes to model stability or performance.

3. The paper specifies the training data used for OneTrackerV2 but does not provide details about the training data or protocols for the compared methods. The authors should clarify whether all models were trained on the same datasets and with comparable data volumes to ensure the fairness and validity of the performance comparisons.

**Questions:**

see Weaknesses

---

### Note · Authors · 2025-11-13

I have read and agree with the venue's withdrawal policy on behalf of myself and my co-authors.